# The Process and Platform for Predicting PM$_{2.5}$ Inhalation and Retention during Exercise

**Hui-Chin Wu [1], Ai-Lun Yang [2], Yue-Shan Chang [3],\*, Yu-Hsiang Chang [3] and Satheesh Abimannan [4]**

[1] Department of Leisure and Sports Management, National Taipei University, Sanxia District, New Taipei City 237303, Taiwan; elaine@gm.ntpu.edu.tw
[2] Institute of Sports Sciences, University of Taipei, Taipei 111, Taiwan; yangailun@gmail.com
[3] Department of Computer Science and Information Engineering, National Taipei University, Sanxia District, New Taipei City 237303, Taiwan; jhangeric@gmail.com
[4] Amity School of Engineering and Technology, Amity University Mumbai, Mumbai 410206, India; satheesha23@gmail.com
\* Correspondence: ysc@mail.ntpu.edu.tw

**Abstract:** In recent years, people have been increasingly concerned about air quality and pollution since a number of studies have proved that air pollution, especially PM$_{2.5}$ (particulate matter), can affect human health drastically. Though the research on air quality prediction has become a mainstream research field, most of the studies focused only on the prediction of urban air quality and pollution. These studies did not predict the actual impact of these pollutants on people. According to the researchers' best knowledge, the amount of polluted air inhaled by people and the amount of polluted air that remains inside their body are two important factors that affect their health. In order to predict the quantity of PM$_{2.5}$ inhaled by people and what they have retained in their body, a process and a platform have been proposed in the current research work. In this research, the experimental process is as follows: (1) First, a personalized PM$_{2.5}$ sensor is designed and developed to sense the quantity of PM$_{2.5}$ around people. (2) Then, the Bruce protocol is applied to collect the information and calculate the relationship between heart rate and air intake under different activities. (3) The amount of PM$_{2.5}$ retained in the body is calculated in this step using the International Commission on Radiological Protection (ICRP) air particle retention formula. (4) Then, a cloud platform is designed to collect people's heart rate under different activities and PM$_{2.5}$ values at respective times. (5) Finally, an APP is developed to show the daily intake of PM$_{2.5}$. The result reveals that the developed app can show a person's daily PM$_{2.5}$ intake and retention in a specific population.

**Keywords:** predicting process; PM$_{2.5}$ inhalation; retention; PM$_{2.5}$; exercise

## 1. Introduction

With the development of the global economy and technological advancements, air pollution has increasingly become a serious concern. According to the report from the WHO [1] and the Environmental Protection Agency (United States) (https://www.epa.gov/pm-pollution/particulate-matter-pm-basics, accessed on 12 September 2021), particulate matter (PM) less than 10 micrometers in diameter (PM$_{10}$) can penetrate deep into human lungs, while PM$_{2.5}$ can penetrate the bronchi and alveoli and can enter the blood stream, which in turn affects the normal functioning of all other organs [2]. A study published in the *Journal of the American Medical Association* [3] showed that PM$_{2.5}$ can cause arterial plaque deposition, which results in vascular inflammation and atherosclerosis. Ultimately, it leads to heart disease or other cardiovascular diseases. Therefore, many global nations have started building a huge number of air quality sensors to sense the quality of air in public places. In order to prevent further pollution of air, more and more studies are being conducted to predict future air quality [4–6]. In addition, the NAPAP (National Acid Precipitation Assessment Program) (https://csl.noaa.gov/aqrsd/reports/napapreport05.

pdf) (accessed on 28 October, 2021) [7] conducts long-term environmental monitoring and reports to the Congress of the US for improving air quality and protecting public health.

Although the research on air quality prediction has become a mainstream research field in recent years, most of the studies conducted in this domain focus on the prediction of urban air quality and pollution. Many traditional machine learning algorithms have been proposed earlier, such as decision trees, GBT (Gradient Boosted Tree Regression), SVR (Support Vector Regression), and deep learning methods, such as LSTM (Long Short-Term Memory Networks), to predict future air quality [4,8–12]. However, these methods did not pay attention to the prediction of the actual impact of these pollutants upon people. As per the researchers' best knowledge, the amount of polluted air inhaled by a person and the amount of air retained in their body are two important factors that affect health. So, it becomes important to predict the amount of $PM_{2.5}$ inhaled and retained by the human body in order to understand the seriousness of the accumulated air pollution and its impact upon the human body.

Maria João Ramos et al. [13] pointed out the calculation approach of the inhaled amount of $PM_{2.5}$. It depends on the $PM_{2.5}$ concentration, the time of exposure under the concentration, and the ventilation amount of the exposed person. Izabela Campos Cozza et al. [14] also proposed another method to calculate the inhaled amount of $PM_{2.5}$. Ramos CA et al. [15] proposed an alternative formula to calculate the inhaled amount of $PM_{2.5}$ [14]. The researchers also considered the weight of the subject when determining the inhaled amount of $PM_{2.5}$. The authors speculated that the inhaled amount of $PM_{2.5}$ varies along with the weight of the subject. According to the study of Zuurbier et al. [16], it is possible to predict the ventilation amount by measuring the heart rate of the subject. Samet et al. [17,18] pointed out that the methods framed so far to determine lung exposure and dose estimation of air pollutants in the epidemiology of air pollution lacked steps to measure a subject's ventilation amount. There is no doubt that outdoor exercise tends to promote physical health [19,20]. Numerous pieces of evidence establish that exercise often has physical and mental health benefits, including an improvement in quality of life and a reduction in morbidity and mortality [21,22]. However, outdoor sports may expose the body to air pollutants such as ozone and nitrogen oxides, which in turn exert a negative impact on the human body [23,24]. This causes physical discomfort such as heart or respiratory tract diseases, lung cancer, and other diseases [25–27].

Therefore, it is important to have a process and platform in place that can estimate and predict the amount of bad air inhaled and retained by the human body so that human health can be improved. Most of the studies [25–28] conducted recently focused on the harm caused by the inhalation of $PM_{2.5}$ by human beings. There is no method available to estimate and predict the amount of $PM_{2.5}$ inhalation and retention by a usual human being. Therefore, the main purpose of this research work is to propose a process to design and implement a platform to estimate and predict $PM_{2.5}$ inhalation and retention values.

During physical exercise, the more intense the exercise is, the faster the heartbeat will be, and accordingly, more air is inhaled. If a person's heart rate is known during normal activities, then it is possible to calculate the relationship between experience and air intake. Accordingly, the air inhaled by the person under normal heart rate can also be calculated. In order to achieve this goal, the following steps were followed. (1) First, a personalized $PM_{2.5}$ sensor was designed and implemented. This sensor can be connected to a mobile phone by the user to sense the $PM_{2.5}$ value around other people. (2) Then, some volunteers were hired to collect their heart rate and air intake by following the Bruce protocol [29] on the treadmill and to calculate the relationship between heart rate and air intake. (3) Based on the values collected, the amount of $PM_{2.5}$ retained in the body was determined as per the International Commission on Radiological Protection (ICRP) [30] air particle retention formula. (4) Fourth, a cloud platform was designed to collect the people's heart rate during different activities and $PM_{2.5}$ value around the people at that time. (5) Finally, an APP was developed and incorporated to show the daily intake of $PM_{2.5}$.

In order to evaluate the process and platform for their efficiency and effectiveness, a SUS questionnaire was constructed to examine the usability of the system. The authors conducted a few experiments to evaluate the latency of the system. The results reveal that the process and platform can be adopted.

The remainder of this paper is organized as follows. Section 2 presents the methods followed to calculate the inhalation amount including the Bruce protocol so as to establish the ventilation amount prediction model, regression analysis of the model, and $PM_{2.5}$ retention amount calculation. Section 3 explains the processes and platform of the system, including the implementation of the data collecting module and the platform and calculation process using hybrid cloud-based system architecture. Section 4 details the results and discussion of the work. Finally, Section 5 presents the concluding remarks and suggestions for future research.

## 2. Methods for Inhalation Amount Calculation

In this section, the method of calculating the amount of air inhalation and retention is explained. The Bruce protocol [29] is exploited in the proposed method, which intends to establish the ventilation amount prediction model on a treadmill to collect the human subject's cardiopulmonary function data (heart rate and ventilation amount).

### 2.1. Bruce Protocol

The Bruce protocol was developed as a step-by-step diagnosis approach by Robert A. Bruce to evaluate the cardiopulmonary function. Tests can also be used to assess the physical condition of athletes. The test intensity used in this work is divided as follows.

Intensity 1. Speed: 2.7 km/h, slope: 10%, lasting 3 min.
Intensity 2. Speed: 4.0 km/h, slope: 12%, lasting 3 min.
Intensity 3. Speed: 5.5 km/h, slope: 14%, lasting 3 min.
Intensity 4. Speed: 6.8 km/h, slope: 16%, lasting 3 min.
Intensity 5. Speed: 8.0 km/h, slope: 18%, lasting 3 min.
Intensity 6. Speed: 8.9 km/h, slope: 20%, lasting 3 min.
Intensity 7. Speed: 9.7 km/h, slope: 22%, lasting 3 min.

The experiment under consideration starts from Intensity 1 to Intensity 7 during when the threshold is set to stop the experiment, considering that the human subject's heart and lung strength may not be able to load so long, as the human subject's heart rate reaches 220. The experiment stops immediately at 85%. If the subject is 20 years old, then the threshold to stop the experiment is at when the heart rate reaches 170 bpm.

This study uses a cardiopulmonary function signal analyzer (VIASYS Vmax Series) and a treadmill as measuring equipment. Before the experiment, the subject wears a mask and ECG patch to collect the information on ventilation amount and heart rate.

### 2.2. Prediction Model of Air Exchange Amount

As per the literature [31], the analysis is performed in this study. An exponential regression model is applied to develop the prediction model for air inhalation. Regression analysis is a statistical analysis method with which one can get to know the existence of a relationship between two or more variables, its direction, and the strength of the correlation. Further, it also helps in establishing a mathematical model to observe specific variables and predict the subject's feelings. Two types of regression analyses exist, such as linear regression and nonlinear linear regression. The exponential regression used in this study belongs to nonlinear linear regression. The formula for the exponential regression model is shown here with Equation (1):

$$Y = ae^{bx} \tag{1}$$

where $e$ is the index (natural exponential function) 2.71828; $a$ and $b$ are unknown parameters; $x$ is the independent variable; and $Y$ is the strain number, which is the predicted result. The method is the same as per the literature [31]. The data used to derive the model and

evaluate the result are collected from recruited volunteers. These volunteers are young students. The average age of male subjects was in the range of 20–39 years, weighing 53–76 Kg, and height in the range of 161–181 cm. Most of them have exercise habits and their weekly exercise time is between 10 and 120 min. The average age of female subjects was in the range of 20–38 years, weighing 43–72 Kg, and height in the range of 152–175 cm. Among the population, three girls do not exercise regularly, whereas the others have exercise habits. Their weekly exercise time is 30–120 min. The heart rate and ventilation amount of 29 subjects were used to establish a predictive model, among which 15 were males and 14 were females. The result is shown in Figure 1a,b, respectively.

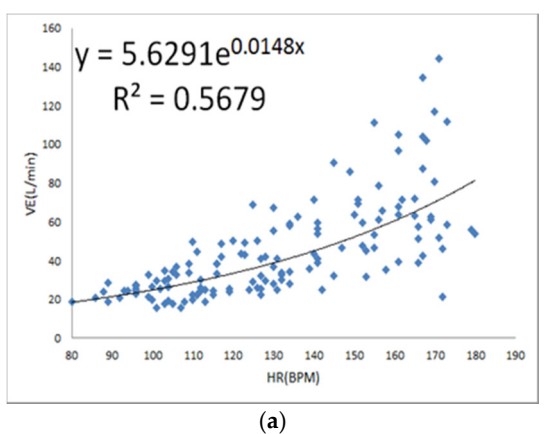
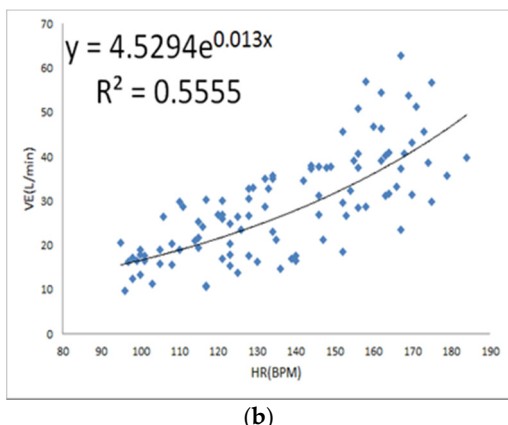

| (**a**) | (**b**) |

**Figure 1.** Prediction model of air exchange amount. (**a**) Male subjects. (**b**) Female subjects.

Here, $R^2$ refers to the coefficient of determination. The definition of $R^2$ represents the ratio of variance of the regression model to all $y_i$ variances. The larger the $R^2$, the greater the proportion of the regression model that can explain the total $y_i$ variance. Therefore, the closer the $R^2$ is to 1.0, the more explanatory power the model will have. The calculation formula of $R^2$ is shown in Equation (2):

$$R^2 = SS_{Reg}/SS_{Total} \tag{2}$$

where $SS_{Reg}$ is the variation of the regression model and $SS_{Total}$ is the variation of the $y_i$ value.

In academic research, the closer the $R^2$ is to 1.0, the better the ideal model value will be. $R^2 = 1$ means that all the observed values are equal to their predicted values (or estimated values). In fact, this situation is rare to achieve. In social science research, $R^2$ values of 0.5 or 0.6 are more commonly achieved, whereas $R^2$ values greater than 0.7 are rare.

### 2.3. Inhalation Amount Calculation

As shown in Figure 1a,b, the ventilation amount prediction formulas have been obtained for both males and females. Only the heart rate data of the subjects need to be obtained. With this value, it is possible to substitute the "x" variable in the ventilation amount prediction formula. If we assume a certain male exerciser "A" with a certain heart rate being 90 bpm, then $5.6291 \times e^{0.0148} \times 90 \fallingdotseq 21$ L. From this calculation, it can be found that when the heart rate of a male exerciser is 90 bpm, then his ventilation amount is about 21 L. If the female exerciser has a heart rate of 90 bpm, then $4.5294 \times e^{0.013} \times 90 \fallingdotseq 15$ L. Finally, the $PM_{2.5}$ inhalation calculation formula from Maria João Ramos et al. [13] and the ICRP retention calculation formula [30] can be used to derive $PM_{2.5}$ retention.

### 2.4. $PM_{2.5}$ Retained Amount Calculation

The breathing mechanism of the human body does not infer that all of the $PM_{2.5}$ is inhaled and is retained during exercise or physical activity. A minuscule amount of $PM_{2.5}$

may be expelled from the body during exhalation. It is a challenge to calculate the amount of $PM_{2.5}$ that may be deposited in the respiratory tract and lungs. The lung deposition formula, proposed by the International Commission on Radiological Protection (ICRP) [30], is shown in Equations (3)–(7).

$$DF = 0.0587 + \frac{0.911}{1 + \exp(4.77 + 1.485 \ln d_p)} + \frac{0.943}{1 + \exp(0.508 - 2.58 \ln d_p)} \tag{3}$$

DF (Deposition Fraction) refers to the amount of total micro-particles deposited in the body, which is simplified by Equations (4)–(7). In the application scenario, DF is about 0.871926254, i.e., about 87% of the $PM_{2.5}$ inhaled each time remains in the body.

$$IF = 1 - 0.5 + \left(1 - \frac{1}{1 + 0.00076\, d_p^{2.8}}\right) \tag{4}$$

where IF (Inhalation Fraction) refers to the inhalation rate and $d_p$ refers to the size of micro-particles. The inhalation rate is calculated according to the size of particles, whereas its result will be different. In the mobile application context, for $d_p$, 2.5 is set as the parameter value. When $d_p$ = 2.5, the IF is about 0.995105109.

$$DF_{HA} = IF\left(\frac{1}{1 + \exp(6.84 + 1.183 \ln d_p)} + \frac{1}{1 + \exp(0.924 - 1.885 \ln d_p)}\right) \tag{5}$$

where $DF_{HA}$ refers to the content of inhaled micro-particles deposited on the head area. When $d_p$ = 2.5, $DF_{HA}$ is about 0.687639644.

$$DF_{TB} = \left(\frac{0.00352}{d_p}\right)\left[\exp\left(-0.234(\ln d_p + 3.40)^2\right) + 63.9 \exp\left(-0.819(\ln d_p - 1.61)^2\right)\right] \tag{6}$$

where $DF_{TB}$ refers to the content of inhaled micro-particles that are deposited in the throat and respiratory tract. When $d_p$ = 2.5, $DF_{TB}$ is about 0.060682591.

$$DF_{AL} = \left(\frac{0.0115}{d_p}\right)\left[\exp\left(-0.416(\ln d_p + 2.84)^2\right) + 19.11 \exp\left(-0.482(\ln d_p - 1.362)^2\right)\right] \tag{7}$$

where $DF_{AL}$ refers to the content of inhaled micro-particles that are deposited in the lungs and alveolar areas. When $d_p$ = 2.5, $DF_{AL}$ is about 0.107680779.

## 3. Processes and Platform of the System

In this work, a user platform is developed and the entire system is based on the ventilation amount prediction model and $PM_{2.5}$ inhalation calculation formula, as discussed earlier. In this section, the implementation of the modules and the system architecture proposed in the work are discussed.

### 3.1. The Implementation of MPS and Collecting Heart Rate

Proof-Of-Concept (POC) [32–34] methodology is the most preferred method in different fields to verify the usability of a system. In this research, the POC method is exploited to verify the usability of the proposed idea. This section presents the implementation of modules to collect the data on $PM_{2.5}$ and heart rate in the proposed platform.

3.1.1. Mobile $PM_{2.5}$ Sensor

First of all, in the hardware development part, the client application uses Android whereas a mobile $PM_{2.5}$ sensor (MPS) is the device used for collecting $PM_{2.5}$ concentration data. This device is developed in the study laboratory. The following devices and modules are used to implement a mobile $PM_{2.5}$ sensor (MPS), which can collect the $PM_{2.5}$ amount anytime, anywhere, and transmit the information to a mobile phone.

MPS was developed in-house by the researchers in this study while commercial PM$_{2.5}$ sensors are avoided for the reasons listed herewith. (1) A PM$_{2.5}$ sensor is not mobile or small enough to be tied on the user's body. (2) These sensors do not allow the data to be either retrieved or stored in the cloud environment developed for this task. So, it is impossible to retrieve the value of PM$_{2.5}$ for calculation at any time. Due to these challenges, the authors developed the MPS by themselves.

(1) *PM$_{2.5}$ sensor module*: The module used in this study can measure particles sized as small as 0.3 μm. It calculates the mass concentration of dust with the following characteristics such as small size, simple wiring, detailed data, and stability.
(2) *Microchip control for PM$_{2.5}$ monitor*: In this work, an open-source single-chip microcontroller, i.e., Arduino Pro Mini, is exploited. This can be used with a Bluetooth communication module, which makes it a good solution for wireless communication.
(3) *Bluetooth transmission module:* In order to transmit the collected PM$_{2.5}$ data to the APP, a Bluetooth transmission module is used to tie within the Arduino microchip. The module supports Bluetooth 2.1 + EDR specification. Therefore, subsequent PM$_{2.5}$ information can be sent to the back-end cloud server and intake calculation can be carried out in the cloud server itself.

### 3.1.2. Connecting Heart Rate Smart Bracelet

A mobile smart bracelet heart rate monitor is used to collect the user's heart rate, their pulse, stress level, and energy fluctuation. The current study made use of a commercial heart rate sensor, i.e., Fitbit, which can connect to a mobile phone via Bluetooth. The dedicated APP in the mobile phone transmits the sensed heart rate data to the back-end public cloud server. A user can see what makes him or her calm, excited, and in between. Through this, the users are free to carry out more activities in line with their physical state. One can also record walking, running, etc. This study uses mainly the heart rate data to calculate the ventilation amount prediction model.

### 3.2. System Platform and Architecture

Next, the system architecture and the operations are introduced in this section. The system exploits hybrid cloud-based deployment [35], as shown in Figure 2. The client APP in the mobile phone serves as the Collaboration Agents [35] to integrate the public cloud and private cloud and is mainly divided into four modules: PM$_{2.5}$ concentration data receiving module, heart rate data acquisition module, PM$_{2.5}$ inhalation calculation module, and historical data module. In addition to the smart bracelet, MPS, and the APP deployed in mobile phones on the front-end, the back-end system uses a hybrid cloud architecture too to deploy the proposed system. The public cloud collects and stores heart rate-related data, whereas the private cloud collects and stores the data regarding PM$_{2.5}$ and the user's personal information.

The whole set of processes and operations of the system are explained herewith in a step-by-step fashion.

1.　The client first turns on Bluetooth and connects it to the mobile PM$_{2.5}$ sensor (MPS).
2.　With MPS, once the client clicks the "Start Exercise" button, the PM$_{2.5}$ data receiving module starts to receive the PM$_{2.5}$ concentration data from MPS.
3.　If the client clicks the "End Exercise" button, it is connected to the smart bracelet first to collect the heart rate data.
4.　After connecting to the heart rate smart bracelet, the user's physiological data are synchronized through the smart bracelet's APP.
5.　After confirming that the physiological data are synchronized, the APP sends the data back to their public cloud.
6.　Then, the authorization of the user and the public cloud is obtained to receive the heart rate data.
7.　After successfully obtaining the authorization, the user's heart rate data are intercepted.

8. The PM$_{2.5}$ data receiving module transmits the concentration data, collected during exercise, to the PM$_{2.5}$ inhalation/retention calculation module.
9. The heart rate data receiving module transmits the collected heart rate data during exercise to the PM$_{2.5}$ inhalation/retention calculation module.
10. Heart rate and PM$_{2.5}$ concentration data are used to calculate PM$_{2.5}$ inhalation and retention, and the results are displayed on the client interface.
11. The calculated results are sent to the historical data module, including heart rate, PM$_{2.5}$ concentration, and PM$_{2.5}$ inhalation and retention data.

The historical data module collects the data from the heart rate data receiving module, PM$_{2.5}$ data receiving module, and PM$_{2.5}$ inhalation/retention calculation module; then it uses RESTful to return it to the private cloud.

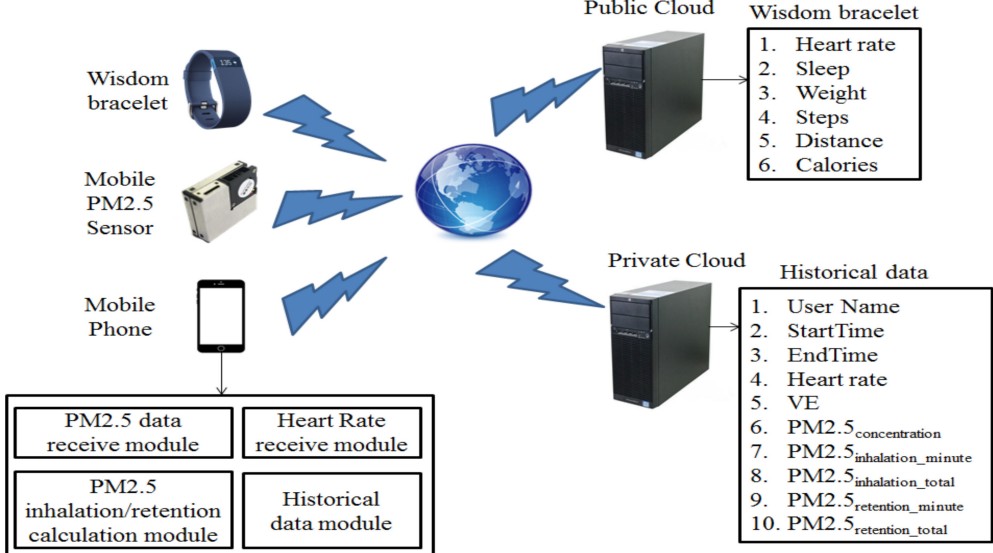

**Figure 2.** Overall architecture—hybrid cloud-based system architecture.

### 3.3. The Module Function and the Calculation Process of Inhalation and Retention

In this section, the module function on the client side and the process to calculate the inhalation and retention are introduced.

#### 3.3.1. PM$_{2.5}$ Concentration Data Receiving Module

This module is primarily responsible for the communication that occurs between the mobile phone and the MPS, and to allow the mobile phone to smoothly receive the concentration data sent by MPS. The steps are explained herewith.

12. Check if the client's Bluetooth function is enabled or not.
13. If it is not turned on, an option pops up to remind the user to turn on the Bluetooth.
14. If the Bluetooth is turned on, it starts to connect with the mobile PM$_{2.5}$ sensor.
15. If the connection is successful, it starts receiving PM$_{2.5}$ concentration data.
16. If the client selects the "End Exercise" button, it ceases the connection with the sensor and turns off the Bluetooth function.
17. If not, continue to step 4 until the "End Exercise" button is selected.

#### 3.3.2. Heart Rate Data Acquisition Module

The main task of this module is to retrieve heart rate data from the public cloud. In this work, heart rate data are stored in Fitbit (https://www.fitbit.com/global/us/about-us, accessed on 25 June 2017).

Therefore, to access the data in the public cloud, a user must register themselves under a user account on the Fitbit website. In order to ensure the security of user data access, the

Fitbit website adopts a well-known OAuth 2.0 (https://oauth.net/2/, accessed on 25 June 2017). OAuth 2.0 is an open standard that allows users to utilize third-party applications for accessing their private resources stored on a website without providing their username and password. The most important thing in this step is to select personal use for the OAuth 2.0 application type registered on the Fitbit website to obtain the detailed information. After completing the registration, Fitbit replies with the following information: *client ID, client secret, client key, authorization URI, access token and request URI.*

To use OAuth 2.0 for accessing personal data on a public website, the client APP must follow the OAuth 2.0 operation process. The instructions are as follows to obtain the authentication code:

(1) Obtain an authentication code: First, the client application needs to obtain an authentication code authorized by the user. The application must first provide *Client_ID*, *Response_Type*, *Redirect_URI*, *Scope* and other parameters and makes a request to the Fitbit server. The sample addresses and parameters are sent as shown in the following weblink: https://www.fitbit.com/ouath2/authorize?response_type=xxx&client_id=xxx&redirect_uri=xxx&scope=xxx accessed on 28 October 2021.

(2) Obtain an access token: After obtaining the authentication code from the public (Fitbit) server, the client application must request an access code. This request is packaged into a RESTful web service with the post method and the following parameters: *Authorization, Content-Type, Code, Grant_Type, Client_ID*, and *Redirect_uri*. Authorization is composed of *client_id:client_secret* format and base 64 encoding. An example of the request is as follows.

> POST https://api.fitbit.com/oauth2/token accessed on 28 October 2021
> Authorization Basic xxx (base 64)
> Content-Type: application/x-www-form-urlencoded
> Body Parameters
> Client_id=xxx&grant_type=authorization_code&redirect_uri=xxx

(3) Acquisition and analysis of the access code: After confirming that the authorization is correct, the authentication server replies with the access code to the user in the form of a json file, as shown below.

> {
> "access_token":"xxx",
> "expires_in":3600,
> "refresh_token":"xxx",
> "token_type":"Bearer",
> "user_id":"xxx"
> }

(4) Obtaining data: Once the access token is obtained, one can obtain data from the server according to the format established by the Fitbit server for the item required. The example is as follows:

> GET https://api.fitbit.com/1/user/-/activities/heart/date/[date]/1d/[detail-level]/time/[start-time]/[end-time].json accessed on 28 October 2021
> Authorization: Bearer xxx

(5) After sending a data retrieval request, the client application receives a json file containing the required data from the Fitbit server, if it is confirmed on the server side, as shown in the example below.

{ "activities-heart-intraday":{"
dataset":[
{"time":"00:00:00","value":64},
{"time":"00:00:05","value":63},
{"time":"00:00:10","value":64},
{"time":"00:00:15","value":65},
{"time":"00:00:20","value":64},
],
"datasetInterval":1,
"datasetType":"second"}
}

### 3.3.3. $PM_{2.5}$ Inhalation/Retention Calculation Module

The main task of this module is to calculate $PM_{2.5}$ intake from the concentration data returned by MPS and the heart rate data obtained from the public cloud. The proposed system uses gender-related ventilation amount prediction model parameters to predict a user's $PM_{2.5}$ inhalation. The ventilation amount conversion formula for men is $5.6291 \times e^{0.0148} \times$ heart rate, and the ventilation amount conversion formula for women is $4.5294 \times e^{0.013} \times$ heart rate. After the ventilation amount is calculated, the $PM_{2.5}$ inhalation formula and the retained amount formula (ICRP) are used to obtain the results.

### 3.3.4. Historical Data Module

The main task of this module is to collect the heart rate data recorded by the heart rate data acquisition module and the $PM_{2.5}$ concentration data recorded by the $PM_{2.5}$ concentration data receiving module and the $PM_{2.5}$ inhalation/retention calculation module. The determined results are then sent to the private cloud through the RESTful web service. The users can make a query to the past sports history records afterwards. MongoDB is used to store the user's exercise history data. The recorded data include user name, start exercise time, end exercise time, heart rate, ventilation amount, $PM_{2.5}$ concentration, $PM_{2.5}$, $PM_{2.5}$ inhaled per minute, the total inhalation, the amount of $PM_{2.5}$ retained per minute, and the total retained amount of $PM_{2.5}$.

## 4. Result and Discussion

### 4.1. Implementation

In this study, an APP was implemented through the Proof-Of-Concept method to present the results. This APP can collect different types of data from the user and present the result. Since this APP is targeted at Taiwanese people, the interface contains traditional Chinese elements, and red fonts are used to present the translated text. When a user starts exercising, the APP will show their current $PM_{2.5}$ value and other data, when starting to record exercise, as shown in Figure 3a.

When the user presses the end exercise button, the system pops up with a dialog window that prompts him or her to use the APP for synchronizing the data immediately. This step is very much required since if the data are not synchronized immediately, it becomes impossible to capture the heart rate data of the user during their exercise period.

After data synchronization is completed, i.e., heart rate data of the user during exercise are sent to the public cloud, the user is then required to log in to their account and authorize the proposed system to retrieve their heart rate data. Finally, the result of the user's exercise, i.e., output, is presented with the time of exercise, $PM_{2.5}$ inhalation, and $PM_{2.5}$ residual distribution data in their body (Figure 3b,c).

When the user clicks "View Exercise History" on the main menu, the system grabs the user's historical exercise data from the private cloud via RESTful and displays their past exercise list. When the user enters a date, it displays the $PM_{2.5}$ inhalation per minute and its retention per minute recorded by the exercise (Figure 3d).

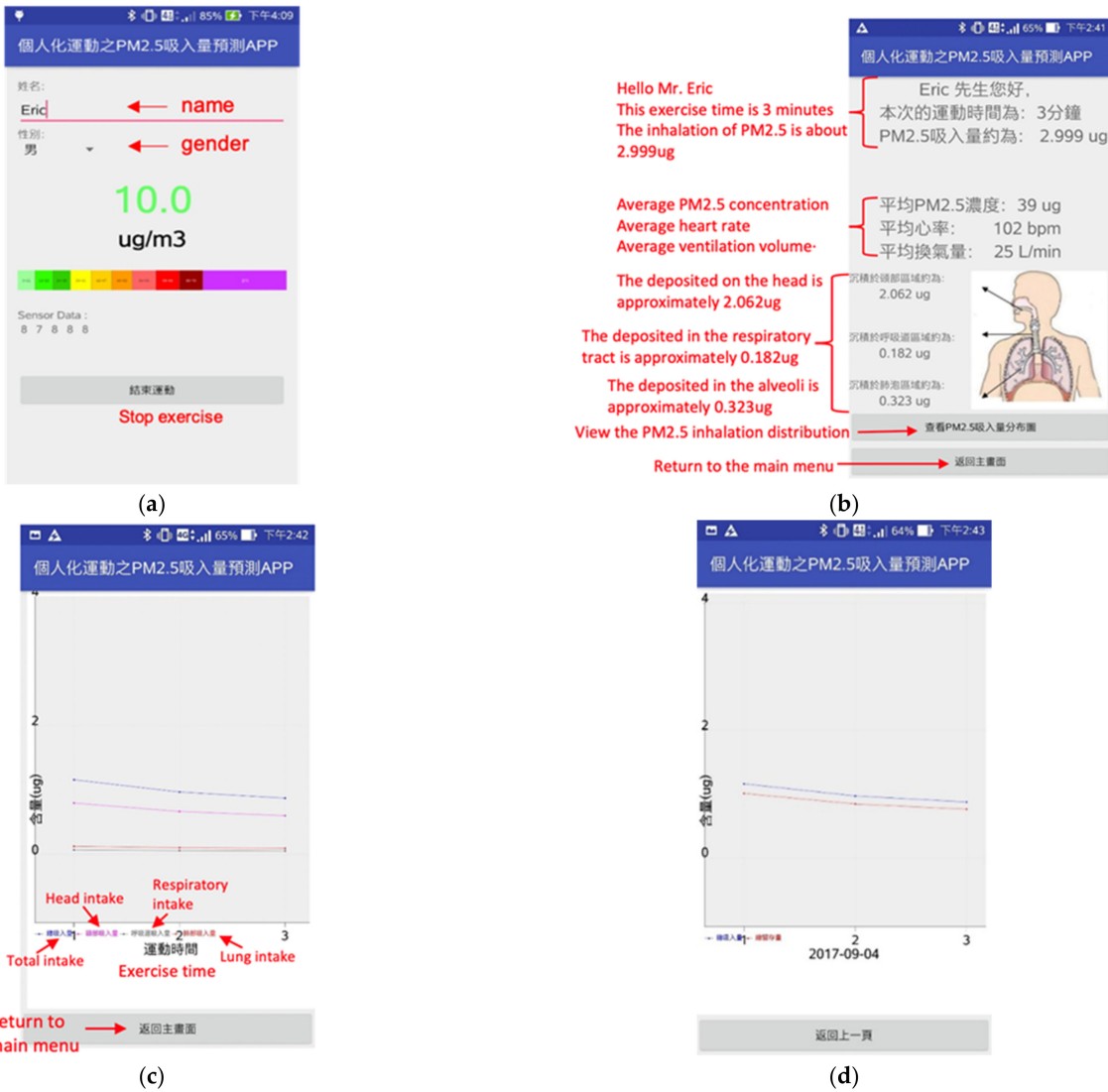

**Figure 3.** The snapshot of implementation result. (**a**) The current PM$_{2.5}$ concentration value displayed once exercise is started; (**b**) The calculation results of PM$_{2.5}$ inhalation/retention; (**c**) Results of PM$_{2.5}$ inhalation per minute; (**d**) The historical record of the exercise (PM$_{2.5}$ inhalation and retention per minute).

*4.2. System Usability Scale (SUS)*

In order to evaluate the practicality of the proposed system, a System Usability Scale (SUS) [36] questionnaire was used to collect the usage results of the users. SUS was developed by John Brooke in 1986, and has been widely used to quickly evaluate product system interfaces, desktop programs, and website interfaces. SUS is recognized as a quick and easy-to-use scoring method. The advantage of using SUS is that the questions are simple and its scores are easy to understand.

4.2.1. SUS Questionnaire

At first, the SUS questionnaire divided the questions on the scale into two groups. Questions such as 1, 3, 5, 7, and 9 are allocated under the first group, while the second group consists of questions 2, 4, 6, 8, and 10. After providing scores to the collected SUS questionnaire, a test score is secured. This score, from the user's comprehensive evaluation, represents the ease of using the system. This score can be used for comparing different systems in terms of ease of use. However, the individual scores for each question remain meaningless and should not be used for comparison or for other purposes. The raw scores for each question should be collected at first. The level, selected by the user for each

question, remains the original score of the question. The raw score is a number between 1 and 5. Subtract 1 from the original score of each question to obtain the number of points to be scored for this question. Then, the number of points to be scored is calculated for each question. Subtract the original score of each question from 5 to obtain the number of points that should be scored for this question. The number of points that should be scored is a number between 0 and 4. Finally, the total score is calculated. Then, add the number of points for each question, and multiply it by 2.5 to obtain the total score. The total score is a number between 0 and 100. The authors searched for 30 users so as to use the system developed in this study. Based on this information, a system practicability survey was conducted. The average number and its standard deviation are shown in Table 1.

**Table 1.** SUS questionnaire analysis results.

| S. No. | Topic | Average | Standard Deviation |
|:---:|:---:|:---:|:---:|
| 1. | I will be willing to use this system often | 3.93 | 0.740 |
| 2. | I think this system is too complicated | 2.30 | 0.837 |
| 3. | I think this system is easy to use | 3.93 | 0.691 |
| 4. | I think I will need the assistance of a technician to use this system | 2.50 | 1.075 |
| 5. | I think the various functions of this system are well integrated with each other | 3.77 | 0.728 |
| 6. | I think there are too many inconsistencies in this system | 2.13 | 0.819 |
| 7. | I can foresee that most people will quickly learn to use this system | 3.80 | 0.925 |
| 8. | I think this system is very difficult to use | 2.17 | 1.053 |
| 9. | I am very confident to be able to use this system | 4.00 | 0.830 |
| 10. | I need to learn a lot of knowledge before I can start using this system | 2.30 | 0.837 |

In order to make SUS scores understandable enough, the authors [36] conducted research on user perception of SUS scores.

4.2.2. SUS Analysis and Results

Here, we present the results of the data acquired from the SUS questionnaire. A total of 30 users participated in the study. Among them, there are seven copies for 0–59 points; five copies for 60–69 points; eight copies for 70–79 points; seven copies for 80–89 points; and three copies for 90–100 points.

In the SUS questionnaire, the positive questions (1, 3, 5, 7, 9) show that the users feel good about the operation of the system and think that it is convenient to use. In terms of inverse questions (questions 2, 4, 6, 8, 10), they show that the users think the system is concise, easy-to-use, and can be operated on their own without much relevant knowledge. In terms of percentiles, the percentage of our system obtaining F (0~59 points) was only about 17.2%. On the whole, users provided a good evaluation for the proposed system.

*4.3. System Execution Time*

In order to evaluate the performance of the proposed system, five different exercise times were measured to evaluate the overall system execution time (as shown in Table 2). They were connected to a $PM_{2.5}$ sensor and received the $PM_{2.5}$ data and public cloud synchronized data. Then, a request was sent to capture the heart rate data and calculate $PM_{2.5}$ inhalation and retention. Among them, the most time-consuming action of the proposed system was "public cloud synchronize data". During this time, the data of the sports bracelet are synchronized back to the public cloud through the mobile phone. However, the most important thing is that they are controlled by the APP. It is impossible to have control over the speed of data return. So, the proposed system exhibited a quick system response time.

**Table 2.** System execution time.

| Test Time (Minute) | Connect PM$_{2.5}$ Sensor and Receive PM2 Data (Seconds) | Public Cloud Synchronized Data (Seconds) | Send Request Grab Heart Rate Data (Seconds) | Calculate PM$_{2.5}$ Inhalation and Retention (Seconds) |
|---|---|---|---|---|
| 7 | 0.721 | 33.360 | 1.125 | 0.022 |
| 18 | 1.518 | 40.140 | 1.280 | 0.039 |
| 30 | 0.996 | 40.133 | 1.128 | 0.056 |
| 38 | 1.188 | 37.656 | 1.243 | 0.413 |
| 62 | 1.082 | 39.209 | 1.260 | 0.076 |
| **Average** | 1.101 | 38.09 | 1.207 | 0.0672 |

*4.4. Users' Perception of PM$_{2.5}$ during Exercise*

In this study, questionnaires were used to collect the data and understand the perception of volunteers towards PM$_{2.5}$ during exercise in the sports field under three dimensions, namely, user's perception about PM$_{2.5}$, user's understanding about PM$_{2.5}$ in the sports field, and user's attention towards PM$_{2.5}$. The result is shown in Table 3. Though most of the people are unaware of air quality, they generally believe that air quality is very important during exercise. Further, the people also think that if there is a device that can let them know PM$_{2.5}$ information in their vicinity, which is important for their health, they will also consider air quality to decide whether to exercise or not. It can be understood from the results and implementation that this model and the app will be helpful for people who exercise.

**Table 3.** Questionnaire analysis of users' perception of PM$_{2.5}$ during exercise.

| Topic | Mean | Standard Deviation | Mean |
|---|---|---|---|
| **User's perception of PM$_{2.5}$** | | | 3.77 |
| 1. I think it is important to understand PM$_{2.5}$ | 4.23 | 0.774 | |
| 2. I know the impact of PM$_{2.5}$ on the human body | 3.80 | 0.887 | |
| 3. I usually care about the amount of PM$_{2.5}$ in my area | 3.27 | 0.980 | |
| **User's understanding of PM$_{2.5}$ in the sports field** | | | 3.41 |
| 4. I know the value of PM$_{2.5}$ in the field during exercise | 3.03 | 1.129 | |
| 5. I think it is important to understand the value of PM$_{2.5}$ in sports fields | 3.73 | 0.828 | |
| 6. When I exercise, I pay attention to the value of PM$_{2.5}$ | 3.33 | 1.124 | |
| 7. I will consider the value of PM$_{2.5}$ in the sports field to decide whether to exercise | 3.37 | 0.999 | |
| 8. I value how much PM$_{2.5}$ I inhale when I exercise | 3.37 | 1.066 | |
| 9. I think it is important to be able to know the PM$_{2.5}$ information near a person at any time | 3.63 | 1.033 | |
| **User's attention to PM$_{2.5}$** | | | 3.75 |
| 10. I value the value of PM$_{2.5}$ in sports fields | 3.60 | 0.932 | |
| 11. There is a device that allows me to know that PM$_{2.5}$ information in my vicinity is important | 3.93 | 0.740 | |
| 12. Personalized PM$_{2.5}$ information at any time is helpful for me to decide whether to exercise | 3.80 | .925 | |
| 13. It is important to know how much PM$_{2.5}$ you inhale at any time during exercise | 3.67 | 0.922 | |

## 5. Conclusions and Future Work

In this research work, a process and a system architecture platform were both designed in such a way that they use mobile devices and wearable devices to collect the user's heart rate and PM$_{2.5}$ concentration in their surrounding environment. The Bruce protocol was applied in this study to find the relationship between heart rate and ventilation amount. Based on the data, exponential regression was exploited to build a prediction model for air exchange amount. The prediction model of heart rate, PM$_{2.5}$ concentration, and ventilation amount was used to calculate the concentration of PM$_{2.5}$ that is inhaled during exercise. Finally, the residual amount of PM$_{2.5}$ in the lung calculation formula was used to calculate the amount of PM$_{2.5}$ retained in the body. A system was built in this study to

quantify the user's PM$_{2.5}$ inhalation/retention amount. This was performed by combining physiological data and environmental data to record the users' PM$_{2.5}$ inhalation and retention content during exercise. This paves the way for people to easily understand the relation between exercise and PM$_{2.5}$. The SUS questionnaire was used to validate the system. The results established the importance of PM$_{2.5}$ for athletes and how far the system developed is helpful to them. In addition, the PM$_{2.5}$ inhalation calculation can also act as a reference indicator for scholars in the medical field. For example, it is important to know the inhaled amount of PM$_{2.5}$ by a person since it affects the body heavily in today's industrialized society.

In future, the user's PM$_{2.5}$ inhalation throughout the day, month, and year should be analyzed to keep a track record on inhalation results. By calculating the accumulated inhalation data over the years, the authors hope to increase the PM$_{2.5}$ inhalation reference directed towards research in epidemiology and medicine.

**Author Contributions:** Conceptualization, H.-C.W. and Y.-S.C.; methodology, H.-C.W. and Y.-S.C.; software, Y.-H.C.; validation, H.-C.W. and S.A.; formal analysis, H.-C.W. and Y.-S.C.; investigation, A.-L.Y. and Y.-S.C.; resources, Y.-S.C.; data curation, Y.-S.C.; writing—original draft preparation, Y.-H.C.; writing—review and editing, Y.-S.C. and S.A.; supervision, A.-L.Y. and Y.-S.C.; project administration, A.-L.Y. and Y.-S.C.; funding acquisition, Y.-S.C. All authors have read and agreed to the published version of the manuscript.

**Funding:** This research was funded by Ministry of Science and Technology, grant number MOST 108-2221-E-305-013-MY3 and The APC was funded by MOST 108-2221-E-305-013-MY3.

**Institutional Review Board Statement:** This work has been approved by the UT-IRB No.: IRB-2016-057 issued by Institutional Review Board of University of Taipei.

**Informed Consent Statement:** Informed consent was obtained from all subjects involved in the study.

**Data Availability Statement:** The data were collected from all subjected involved in the study.

**Acknowledgments:** This work was partially supported by the Ministry of Science and Technology of Taiwan, Republic of China under Grant No. MOST 109-2119-M-305-001-A and MOST 108-2221-E-305-013-MY3, and by National Taipei University under Grant No. 108-NTPU_A-H&E-143-001.

**Conflicts of Interest:** The authors declare no conflict of interest.

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
