# Peer review of "The Process and Platform for Predicting PM2.5 Inhalation and Retention during Exercise"

_processes, doi:10.3390/pr9112026_

Round 1

Reviewer 1 Report

The English grammar needs to be substantially improved, including in the abstract.  

How does the sensor developed compare to others already commercially available across the Globe.  If this was presented then it needs to be expanded upon.  

The references contain some US references that do not appear to be directly relevant, but will let it go for now.

Clarify the details behind the data used to test the algorithms of this device. For example, Where do they live, What is their age, and lifestyles?  This could then enhance the discussion and help advance the understanding.  If they are obtained from one of the references used, ok, say so.  

Author Response

Thank you for the reviewer’s comment. I have fixed all the issues highlighted by the Reviewer 1, as follows:

  1. The English grammar needs to be substantially improved, including in the abstract.

Response:Thank you for the comment. I have asked the native speaker to polish the English of the paper thoroughly. I think that the current version has fixed all grammar errors and mistakes.

  1. How does the sensor developed compare to others already commercially available across the Globe. If this was presented then it needs to be expanded upon. 

Response:Thank you for the comment. We have added a paragraph into Section 3.1.1 to address the concern, as follows. This reason to develop MPS ourselves is that before the start of this work, the commercial PM2.5 sensors may share one of following problems: 1. PM2.5 sensor is not mobile or small enough for users to be tied on the body. 2. The data measured by the sensor cannot allow us to retrieve and store on the cloud environment we have built, so that it is impossible for us to retrieve the value of PM2.5 for calculation at any time. Therefore, we need to develop it ourselves.

  1. The references contain some US references that do not appear to be directly relevant, but will let it go for now.

Response:Thank you for the comment. We have examined and arranged all references thoroughly

  1. Clarify the details behind the data used to test the algorithms of this device. For example, where do they live, What is their age, and lifestyles? This could then enhance the discussion and help advance the understanding.  If they are obtained from one of the references used, ok, say so. 

Response:Thank you for the valuable comment. We have added a paragraph into the Section 2.2, as follows:

The data used to derive the model and evaluate the result are collected from recruited volunteers. These volunteers are young students. The age of male subjects are 20-39 years old, weigh 53-76Kg, and height 161-181cm. Most of them have exercise habits and the weekly exercise time is 10-120 minutes. The age of female subjects are 20-38 years old, weigh 43-72Kg, and height 152-175cm. There are three girls who do not usually exercise, and others have exercise habits. The weekly exercise time is 30-120 minutes.

Reviewer 2 Report

  • The writing in the abstract and introduction should be revised. There are some grammar and punctuation mistakes.
  • The introduction is a bit long. Some parts are repeated (e.g., steps for the goals lines 57-66 and contributions of the paper lines 72-79), and the background and related work should be added to the introduction before the goals. In addition, there is no need to include so much detail about other research (that should be synthesized) and point out which of the equations you are using and why. 
  • The paper is way too long; some explanations should not be in the paper because they are basic math (like the regression model explanation, the coefficient of determination, and so on). If you want, you can add them in a supplementary information section. 
  • Figures 2 and 3 can be combined. 
  • In section 3.3, lines 281-286 can be removed, along with lines 323-329. That could be in an annex for supplementary information.
  • There are too many figures. Some of them can be combined, such as figures 4, 5, and 6. Figures 7, 8 and 9 can be removed because you already explained them in the text. Lines 450-503 should go to supplementary information.
  • Figures 10-13 should be combined too.
  • I believe figures 14-16 are not necessary.

Author Response

Thank you for the reviewer’s comment. I have fixed all the issues highlighted by the Reviewer 2, as follows:

  1. The writing in the abstract and introduction should be revised. There are some grammar and punctuation mistakes.

Response:Thank you for the comment. I have asked the native speaker to polish the English of the paper thoroughly. I think that the current version has fixed all grammar errors and mistakes.

  1. The introduction is a bit long. Some parts are repeated (e.g., steps for the goals lines 57-66 and contributions of the paper lines 72-79), and the background and related work should be added to the introduction before the goals. In addition, there is no need to include so much detail about other research (that should be synthesized) and point out which of the equations you are using and why.

Response: Thank you for the comments. We have removed lines 72-79. We have also added the main parts of Background and Related work before the goals of Section 1, and also deleted most of the unnecessary content.

  1. The paper is way too long; some explanations should not be in the paper because they are basic math (like the regression model explanation, the coefficient of determination, and so on). If you want, you can add them in a supplementary information section.

Response: Thank you for comment. I have examined all parts and deleted them mentioned in the Reviewer.

  1. Figures 2 and 3 can be combined.

Response: Thank you for the suggestion. We have combined these two figures into Figure 2.

  1. In section 3.3, lines 281-286 can be removed, along with lines 323-329. That could be in an annex for supplementary information.

Response: Thank you for your comment. We have removed the examples. (Line 281-286 and 323-329)

  1. There are too many figures. Some of them can be combined, such as figures 4, 5, and 6. Figures 7, 8 and 9 can be removed because you already explained them in the text. Lines 450-503 should go to supplementary information.

Response: Thank you for the suggestions. We have removed the Figures 4 and 5; Figure 7, 8, and 9; and removed the Lines 450-503.

  1. Figures 10-13 should be combined too.

Response: Thank you for the comment. We have combined them into Figure 4.

  1. I believe figures 14-16 are not necessary.

Response: Thank you for the comment. We have removed Figure 14-16.

Round 2

Reviewer 1 Report

In Abstract, 'these pollution' needs to be re-written to, "these pollutants". There are other like issues through out. 

In Paper, Simply wanted the responses in the text.  You have younger and newer readers to this field.  They do not likely understand  your message.  No reference to NAPAP and like literature is not ideal, and would have made this paper more useful to this community. 

Author Response

  1. In Abstract, 'these pollution' needs to be re-written to, "these pollutants". There are other like issues through out. 

Response: Thank you of the comment. I have revised the point.

  1. In Paper, Simply, wanted the responses in the text. You have younger and newer readers to this field. They do not likely understand your message. No reference to NAPAP and like literature is not ideal, and would have made this paper more useful to this community. 

Response: Thank you for the comments. We have found the relevant document of NAPAP and added them appropriately to the first section of this paper to enrich the references and the readability of the article. This research exploits the Internet Of the Thing (IOT), Information Technology, and Cloud computing platform to predict PM2.5 inhalation during exercise, so that the relationship between PM2.5 inhalation and physical health can be explored in the future. I think that it is a multi-disciplines study, including the air pollution, information technology, and sports. For young and newer readers, they need to have some IT technology concepts and know the human body's reaction during exercise. I believe it is easy to understand the method we proposed and know that the value of the proposed method.

Reviewer 2 Report

Authors have corrected everything that I recommended. The paper looks much better now. The coherence and transitions in the paragraph in the lines 49-62 can be improved, but the rest looks ok. 

Author Response

Reviewer #2 comments

  1. Authors have corrected everything that I recommended. The paper looks much better now. The coherence and transitions in the paragraph in the lines 49-62 can be improved, but the rest looks ok.

Response: Thank you for the comment. I have revised the paragraph in the line 49-62.
